# Association between physical activity levels and functional recovery, quality of life, and psychological well-being in patients undergoing physiotherapy for musculoskeletal disorders: A cross-sectional study

Batool Abdulelah Alkhamis[1], Ravi Shankar Reddy [1*], Mastour Saeed Alshahrani[1], Zuhair Al Salim[2], Faisal M. Alyazedi[3], Ahmed Mohamed Elshiwi[4], Ghada Mohamed Koura[1], Devika Rani Sangadala[1], Debjani Mukherjee [1], Saeed Y. Al Adal[5], Hussain Saleh H. Ghulam[5]

1 Program of Physical Therapy, Department of Medical Rehabilitation Sciences, College of Applied Medical Sciences, King Khalid University, Abha, Saudi Arabia, 2 Department of Sport Science and Physical Activity, University of Hafr Al Batin, Hafar Al Batin, Saudi Arabia, 3 Physical Therapy Department, Prince Sultan Military College of Health Sciences, Dhahran, Saudi Arabia, 4 Department of Physiotherapy, Saudi German Hospital, Abha, Aseer, Saudi Arabia, 5 Department of Medical Rehabilitation Sciences – Physiotherapy Program, College of Applied Medical Sciences, Najran University, Najran, Saudi Arabia

* rshankar@kku.edu.sa

## Abstract

Physical activity is known to enhance functional recovery and quality of life (QoL) in individuals with musculoskeletal disorders; however, the extent to which varying levels of physical activity influence rehabilitation outcomes—and how demographic and clinical factors moderate these effects—remains inadequately understood. This cross-sectional study investigated the associations between physical activity levels and functional recovery, QoL, and psychological well-being among 286 adults undergoing physiotherapy at a tertiary care hospital in Abha, Saudi Arabia. Participants were classified into low, moderate, or high physical activity groups based on metabolic equivalent of task (MET) hours per week, using the culturally adapted Arabic version of the International Physical Activity Questionnaire – Short Form (IPAQ-SF). Outcome measures included the Patient-Specific Functional Scale (PSFS), the 36-Item Short Form Health Survey (SF-36), and the Hospital Anxiety and Depression Scale (HADS). Multiple linear regression analyses, adjusted for age, gender, socioeconomic status, and comorbidities, revealed that higher physical activity levels were significantly associated with greater functional recovery ($\beta = 0.36$, 95% CI: 0.06 to 0.66, $p = 0.021$), higher QoL ($\beta = 0.42$, 95% CI: 0.10 to 0.74, $p = 0.011$), and lower anxiety ($r = -0.41$, $p = 0.013$) and depression scores ($r = -0.38$, $p = 0.022$). Moderation analysis indicated that age and comorbidities negatively influenced the relationship between physical activity and recovery outcomes. The full model explained 68% of the variance in functional recovery ($R^2 = 0.68$, $\Delta R^2 = 0.16$ with moderators, $p = 0.023$).

**Data availability statement:** The datasets generated and analyzed during the current study are publicly available in the Zenodo repository at https://doi.org/10.5281/zenodo.14162315.

**Funding:** The authors extend their appreciation to the Deanship of Research and Graduate Studies at King Khalid University, KSA, for funding this work through a large research group under grant number RGP. 2/548/46. The funders had no role in the study design, data collection and analysis, decision to publish, or preparation of the manuscript.

**Competing interests:** The authors have declared that no competing interests exist.

These findings indicate that higher physical activity levels are associated with more favorable rehabilitation outcomes among individuals undergoing physiotherapy for musculoskeletal disorders; however, due to the cross-sectional design, causality cannot be inferred, and the generalizability of these associations requires further investigation.

## Introduction

Physical activity is a core component of musculoskeletal rehabilitation, contributing to improved joint mobility, muscular strength, and endurance [1]. These factors are essential for enhancing functional recovery and health-related quality of life (QoL) in patients undergoing physiotherapy for musculoskeletal disorders [2]. Beyond physical improvements, physical activity has been shown to positively influence mental health by alleviating symptoms of anxiety and depression and enhancing overall psychological well-being [3]. Although the general benefits of physical activity are well-documented, the specific impact of varying activity levels on recovery outcomes within physiotherapy settings remains insufficiently explored [4].

Emerging evidence suggests a dose-dependent relationship between physical activity and health outcomes; however, this association may be moderated by demographic and clinical factors such as age, gender, socioeconomic status, and comorbidities [5]. These variables can influence both the capacity to engage in physical activity and the extent to which it yields therapeutic benefits [4].

This study examines the associations between physical activity levels and functional recovery, QoL, and psychological well-being in individuals undergoing physiotherapy for musculoskeletal conditions. Additionally, it investigates whether age, gender, socioeconomic status, and comorbidities moderate these relationships. It is hypothesized that higher physical activity levels will be associated with better outcomes, though the strength of these associations may vary across subgroups.

## Methods

### Study setting, design, and ethics

The study was a cross-sectional analysis conducted at the Department of Physical Medicine and Rehabilitation at KKU Hospital, a tertiary care hospital specializing in musculoskeletal and rehabilitation services. The study was conducted from 13 July 2024–18 March 2025, with Institutional Review Board approval from KKU Hospital (REC# 2024–6743) to ensure adherence to ethical standards. The extended data collection period was necessary to ensure recruitment of a representative sample across all physical activity levels and to accommodate fluctuations in patient inflow across outpatient departments. This also allowed for balanced stratification across demographic subgroups relevant to the moderation analysis. Before enrollment, all participants provided written informed consent, and the study procedures complied with the ethical principles of the Declaration of Helsinki.

## Participants

This study recruited participants from the outpatient physiotherapy department of KKU Hospital, where individuals with musculoskeletal disorders were receiving rehabilitation. The clinical environment included general and specialty musculoskeletal physiotherapy clinics within a tertiary hospital setting, offering services such as manual therapy, electrotherapy, and individualized exercise-based rehabilitation. Recruitment was conducted via direct referral from treating physiotherapists during routine clinical visits. Eligible patients were approached in private consultation rooms, and confidentiality was maintained throughout consent and data collection procedures. Personal identifiers were removed from all datasets prior to analysis. Diagnostic criteria included clinically confirmed musculoskeletal conditions affecting mobility and function, as documented by a licensed physician based on physical examinations, imaging studies, and patient-reported symptoms [12]. Common diagnoses among participants included osteoarthritis, lower back pain, and joint injuries, which significantly impaired functional ability and required structured physiotherapy interventions [13]. The distribution of diagnoses in the sample was as follows: chronic lower back pain (38.46%), osteoarthritis (29.02%), shoulder or knee joint injuries (22.73%), and other musculoskeletal conditions such as tendinopathies or post-fracture rehabilitation (9.79%). Diagnosis was based on clinical examination and imaging, where applicable, as documented in participants' medical records. The inclusion criteria for this study were as follows: adults aged 18 years and older who had been diagnosed with a musculoskeletal disorder requiring physiotherapy, with an estimated recovery time of at least eight weeks. Participants were also required to have the cognitive capacity to understand and complete self-reported assessments on QoL, physical activity, and psychological well-being. Exclusion criteria included patients with acute infections, recent fractures, or neurological disorders that could interfere with physiotherapy outcomes. Individuals with severe mental health conditions, such as major depressive disorder or psychosis, which might limit adherence to the rehabilitation protocol or influence self-report accuracy, were also excluded. Patients who were non-ambulatory or had contraindications to physical activity, such as unstable cardiovascular disease, were not eligible.

Participants were selected through purposive sampling based on study eligibility criteria. No follow-up assessments were conducted; all data were collected at a single time point consistent with the cross-sectional design. A total of 412 patients were initially screened for eligibility during the recruitment period. Of these, 286 met the inclusion criteria and consented to participate, while 126 were excluded. Exclusions included 53 patients who did not meet diagnostic or cognitive inclusion criteria, 41 who declined participation, and 32 who were unable to commit to the assessment schedule. A participant flow diagram has been provided (Fig 1) to illustrate the recruitment process, including numbers assessed for eligibility, excluded participants (with reasons), and those included in the final analysis, as recommended by STROBE. The final enrolled cohort thus represented 69.4% of those screened. While purposive sampling enabled targeted recruitment of individuals receiving active physiotherapy for musculoskeletal disorders, this approach may limit generalizability

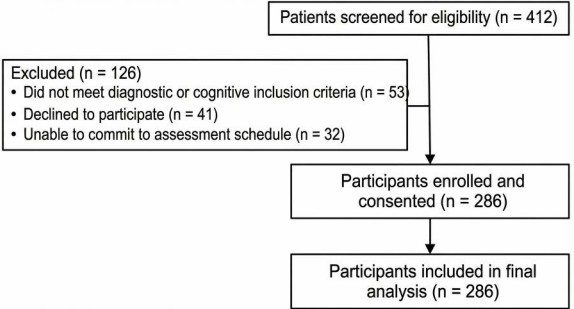

**Fig 1. Participant flow diagram for recruitment and inclusion.**

and introduce selection bias due to its reliance on clinical judgment and participant availability. These limitations should be considered when interpreting the findings. Their physiotherapists approached eligible individuals and provided detailed information about the study's purpose, procedures, and role. This recruitment process aimed to recruit a diverse sample that reflects typical patients undergoing physiotherapy for musculoskeletal disorders, ensuring enough variability in physical activity levels and other demographic characteristics relevant to the study's objectives.

## Variables

The primary outcome variables in this study included functional recovery, QoL, and psychological well-being. Functional recovery was evaluated using the Patient-Specific Functional Scale [6], where participants rated their limitations in daily activities on a scale from 0 to 10, with higher scores representing greater functional capacity. Participants were instructed to identify three to five daily activities they found difficult due to their musculoskeletal condition. Each activity was rated from 0 (unable to perform) to 10 (able to perform at prior level). The final PSFS score represented the mean of all selected activities per participant, following standard PSFS scoring recommendations. The number of selected activities was recorded to ensure consistency in score interpretation. QoL was assessed using the Short Form-36 Health Survey (SF-36) [15], a validated instrument comprising eight subscales that cover physical functioning, physical role limitations, bodily pain, general health, vitality, social functioning, emotional role limitations, and mental health [7]. Higher scores on each subscale and overall indicate better perceived QoL [7]. Psychological well-being was measured through the Hospital Anxiety and Depression Scale (HADS) [8], a 14-item tool that separately measures anxiety and depression levels, with higher scores reflecting greater psychological distress [8].

The primary independent variable was physical activity level, measured in metabolic equivalent of task (MET) hours per week [17]. Participants completed the Arabic-translated IPAQ-SF, which records physical activity in MET-minutes per week. These scores were computed using standard IPAQ scoring protocols and subsequently converted to MET-hours per week by dividing total MET-minutes by 60, to enhance interpretability in regression models [17]. The physical activity data were collected using the short form of the International Physical Activity Questionnaire (IPAQ-SF), which has been translated and culturally validated for Arabic-speaking populations. The IPAQ-SF is a widely used tool for estimating MET-minutes per week across walking, moderate, and vigorous activities, and was scored according to standard IPAQ protocols. Based on this measure, participants were categorized into three groups: low, moderate, and high physical activity. The classification of physical activity levels was based on the official IPAQ-SF scoring protocol. Participants with less than 600 MET-min/week were categorized as having low physical activity, those with 600–2999 MET-min/week as moderate, and those with 3000 MET-min/week or more as high physical activity. These thresholds are consistent with WHO recommendations and have been validated for use in adult populations across different cultural contexts. In the present sample, 27.97% of participants were categorized as low activity, 42.31% as moderate, and 29.72% as high activity, ensuring a balanced distribution that supports group comparisons. Demographic and lifestyle variables, such as age, gender, socioeconomic status, and comorbidities, were also collected to examine their potential moderating effects on the primary outcomes [18]. Baseline severity was operationalized as the initial functional recovery score reported by each participant using the Patient-Specific Functional Scale (PSFS), where lower scores at intake indicated more severe functional impairment. Socioeconomic status was classified as low, middle, or high based on a composite of two criteria: (1) highest level of education attained and (2) self-reported monthly household income. High socioeconomic status was defined as having a college degree or higher and a monthly income above the national median (as per Saudi General Authority for Statistics, 2024 figures [9]). Medication use was recorded as a binary variable (yes/no) based on medical records and self-reports, encompassing any prescribed medications for chronic conditions relevant to musculoskeletal health (e.g., analgesics, nonsteroidal anti-inflammatory drugs, antihypertensives, or diabetes medications). Comorbidities were similarly coded as a binary variable (present/absent) and included physician-confirmed diagnoses of cardiovascular disease, diabetes, or obesity. These were verified through patient medical records and used both as covariates and moderators in

the regression models to assess their interaction with physical activity on recovery outcomes. Age was recorded in years, while gender was categorized as male or female. Socioeconomic status was determined based on a combination of education level and self-reported income and categorized into low, middle, or high socioeconomic status. Comorbidities, including cardiovascular disease, diabetes, and obesity, were recorded based on participants' medical records and self-reports, with a binary variable indicating the presence or absence of any comorbidity [19]. Smoking status and employment status were likewise collected as categorical variables, with smoking categorized as current smoker or non-smoker, and employment as employed or unemployed.

## Data collection process

Data were collected through a combination of patient self-reports, medical records, and standardized assessments, ensuring the use of validated and reliable tools for each outcome and independent variable. This methodological approach facilitated a comprehensive examination of the relationships between physical activity levels and functional recovery, QoL, and psychological well-being, with adjustments for potential confounding variables. To ensure data accuracy and minimize potential biases, several rigorous data collection procedures were implemented. To further address potential sources of bias, blinding was implemented during data entry and analysis, whereby research assistants were unaware of participants' physical activity groupings. Additionally, validated instruments were used for all primary outcomes to reduce measurement bias. Selection bias was minimized through purposive sampling across multiple clinics within the hospital, and the inclusion of diverse musculoskeletal diagnoses enhanced external validity. Data collectors, including research assistants and physiotherapists involved in the study, underwent specialized training before the commencement of data collection. This training covered standardized protocols for administering questionnaires, recording demographic and clinical information, and addressing common sources of error in self-reported measures. To minimize recall bias, participants were encouraged to reflect on recent activity patterns during questionnaire completion, in accordance with IPAQ-SF instructions. In addition, where feasible, self-reported clinical data were cross verified with participants' medical records to confirm the presence of comorbidities and validate baseline health status. This cross-verification process ensured consistency between self-reports and documented clinical data, enhancing the accuracy of key variables used in the analysis. While the study relied primarily on self-reported scales, blinded data entry and scoring procedures were implemented. Baseline health status was operationalized as a composite clinical assessment score based on the physiotherapist's initial evaluation, incorporating range of motion limitations, pain interference during daily activities, and overall physical function. Scores were recorded on a 0–10 scale, with higher scores indicating greater impairment. This score was extracted from standardized intake records and used as a covariate in the regression models.

## Sample size calculation

An a priori sample size was determined using G*Power software (version 3.1) to ensure sufficient statistical power for detecting significant effects. Based on effect sizes from similar studies examining how physical activity influences functional recovery and QoL in musculoskeletal rehabilitation, a medium effect size ($f^2 = 0.15$) was chosen for the primary analyses. With a significance level of $\alpha = 0.05$, a target power of 80%, and a model including up to 10 predictor variables (covering physical activity levels, demographic factors, and clinical covariates), the calculation indicated a minimum required sample size of 286 participants.

## Statistical analysis

Data were analyzed using SPSS software (Version 24.0). Prior to model interpretation, assumptions for linear regression were evaluated. The normality of residuals was confirmed via visual inspection of Q–Q plots and Shapiro–Wilk tests ($p > 0.05$), while homoscedasticity and linearity were assessed using residuals versus fitted value plots. No major

violations of these assumptions were identified. Multicollinearity diagnostics showed variance inflation factor (VIF) values ranging from 1.04 to 1.28, indicating low multicollinearity and stable model estimation. Associations between physical activity levels and the primary outcomes—functional recovery, quality of life (QoL), and psychological well-being—were examined using Pearson correlation analyses and multiple linear regression models. Physical activity served as the main independent variable, with each outcome assessed in a separate regression model. Covariates were selected a priori based on theoretical relevance and prior evidence. For all models, age, gender, socioeconomic status, and presence of comorbidities were included. Additional covariates—BMI, medication use, smoking status, and employment status—were incorporated into models examining QoL and psychological outcomes due to their known influence on health-related variables. These covariates, described in Table 1, were entered hierarchically to evaluate their individual and combined effects, enhancing the accuracy of physical activity estimates.

Missing data were assessed for each variable and found to be < 5% overall. As the missingness appeared completely at random (MCAR), listwise deletion was employed to maintain analytical consistency while avoiding the assumptions associated with data imputation. Sensitivity analyses excluding these cases yielded results consistent with the main models. Group differences in functional recovery scores across physical activity levels (low, moderate, high) were evaluated using one-way ANOVA. Significant effects were further examined using Tukey's post hoc tests, and effect sizes ($\eta^2$) were calculated to assess magnitude. For QoL and psychological outcomes, ANCOVA was used with physical activity group as the independent factor and relevant covariates included. The ANCOVA for QoL showed a significant main effect of physical activity, $F_{(2, 281)} = 5.23$, $p = 0.006$, partial $\eta^2 = 0.27$. Adjusted mean SF-36 scores were 58.76 (SD = 14.21) for the low activity group, 66.04 (SD = 13.56) for moderate, and 70.91 (SD = 12.67) for high activity participants. Bonferroni-adjusted post hoc comparisons indicated significant differences between the low and high groups (p = 0.004, 95% CI [2.45, 11.96]) and between low and moderate groups (p = 0.031, 95% CI [0.45, 10.01]).

To examine potential moderating effects, hierarchical regression models were extended to include interaction terms (e.g., physical activity × age, × gender, × socioeconomic status). Changes in explained variance ($\Delta R^2$) were used to assess the contribution of these interaction terms, providing insight into whether the strength of association varied across subgroups.

**Table 1. Participant Demographics and Clinical Profiles.**

| Characteristic | Mean ± SD/ % |
|---|---|
| Age (years) | 45.32 ± 12.56 |
| Gender (Male, %) | 53.42% |
| BMI (kg/m²) | 26.87 ± 04.23 |
| Duration of Condition (months) | 14.56 ± 08.32 |
| Physical Activity Level (MET hours/week) | 21.43 ± 10.56 |
| Functional Recovery Score (PSFS) | 06.89 ± 02.34 |
| Quality of Life (SF-36 score) | 65.23 ± 15.87 |
| Anxiety (HADS score) | 08.45 ± 03.12 |
| Depression (HADS score) | 07.23 ± 02.56 |
| Socioeconomic Status (Low, %) | 35.23% |
| Comorbidities (%, any) | 42.12% |
| Smoking Status (Smoker, %) | 27.54% |
| Education Level (College Degree, %) | 61.23% |
| Employment Status (Employed, %) | 72.54% |

BMI, Body Mass Index; MET, Metabolic Equivalent of Task; PSFS, Patient-Specific Functional Scale; SF-36, Short Form Health Survey; HADS, Hospital Anxiety and Depression Scale.

## Results

The demographic and clinical characteristics of the participants revealed a mean age of 45.32 years and a nearly equal gender distribution with 53.42% males (Table 1). Participants presented a mean BMI of 26.87 kg/m², reflecting a moderate range. The average duration of the musculoskeletal condition was 14.56 months. In terms of physical activity, the mean level was 21.43 MET hours per week, and functional recovery, as measured by the PSFS, averaged 6.89. QoL, as assessed by the SF-36, showed a mean score of 65.23. Anxiety and depression levels, as measured by the Hospital Anxiety and Depression Scale (HADS), were 8.45 and 7.23, respectively. Socioeconomic status indicated that 35.23% of participants were classified as low socioeconomic status, while 42.12% reported comorbidities. Smoking was prevalent in 27.54% of the cohort, with 61.23% having attained a college degree and 72.54% being employed.

A moderate positive correlation was identified between physical activity levels and functional recovery outcomes, suggesting that individuals reporting higher levels of physical activity tended to report greater functional capacity ($r = 0.45$, $p = 0.013$). In multiple linear regression analysis, physical activity remained a significant independent predictor of functional recovery ($\beta = 0.38$, $p = 0.008$), consistent with the model shown in Table 2 and Fig 2. Notably, both age ($\beta = -0.18$, $p = 0.045$) and baseline severity ($\beta = -0.29$, $p = 0.012$) demonstrated significant negative associations with recovery outcomes, suggesting reduced improvement in older individuals and those with more severe initial impairments.

Significant group differences in functional recovery were identified across physical activity levels, with the high activity group demonstrating the greatest improvement compared to both moderate and low activity groups (Fig 3). The one-way ANOVA yielded a statistically significant result, $F(2, 283) = 4.57$, $p = 0.034$, with a moderate effect size ($\eta^2 = 0.24$). Tukey's post-hoc test revealed that the high physical activity group had significantly higher functional recovery scores ($M = 7.45$, $SD = 1.98$) compared to the low activity group ($M = 6.12$, $SD = 2.43$), with a mean difference of 1.33 (95% CI [0.48, 2.18], $p = 0.009$). No significant difference was observed between moderate and high activity groups ($p = 0.124$). Post-hoc analysis confirmed that the difference between the low and high physical activity groups was statistically significant ($p < 0.05$), suggesting a graded association between physical activity levels and functional recovery outcomes, although this trend should be interpreted as observational rather than causal.

Higher physical activity levels were significantly associated with improved QoL and psychological well-being, as indicated by a positive beta coefficient ($\beta = 0.38$, $p = 0.008$), after adjusting for multiple covariates (Table 2). Notably, baseline

**Table 2. Multiple regression and pearson correlation: influence of physical activity on quality of life and psychological well-being.**

| Variable | Beta Coefficient (β) | Standard Error (SE) | 95% Confidence Interval | VIF | p-value |
|---|---|---|---|---|---|
| Physical Activity Level | 0.38 | 0.06 | 0.26 to 0.50 | 1.22 | .008 |
| Age | −0.14 | 0.04 | −0.22 to −0.06 | 1.15 | .029 |
| Gender (Male = 1) | 0.06 | 0.03 | 0.00 to 0.12 | 1.04 | .064 |
| Baseline Health Status | −0.31 | 0.05 | −0.41 to −0.21 | 1.28 | .005 |
| Medication Use (Yes = 1) | −0.11 | 0.04 | −0.19 to −0.03 | 1.18 | .032 |
| BMI (kg/m²) | −0.09 | 0.03 | −0.15 to −0.03 | 1.13 | .024 |
| Socioeconomic Status (High = 1) | 0.17 | 0.05 | 0.07 to 0.27 | 1.20 | .014 |
| Smoking Status (Smoker = 1) | −0.13 | 0.05 | −0.23 to −0.03 | 1.19 | .021 |
| Comorbidities (Present = 1) | −0.22 | 0.06 | −0.34 to −0.10 | 1.26 | .011 |
| Employment Status (Employed = 1) | 0.15 | 0.05 | 0.05 to 0.25 | 1.11 | .019 |
| Constant | 4.43 | 0.72 | 3.01 to 5.85 | – | .001 |

β; Beta Coefficient; SE; Standard Error; VIF; Variance Inflation Factor; BMI; Body Mass Index.

Standardized beta coefficients (β) are reported to indicate effect sizes for all regression models. For ANCOVA analyses, partial eta squared (η²) values are provided to quantify the magnitude of between-group differences. All models met the necessary assumptions for linear regression, and no violations were observed in residual diagnostics.

health status (β = −0.31, *p* = 0.005), presence of comorbidities (β = −0.22, *p* = 0.011), and smoking (β = −0.13, *p* = 0.021) were independently associated with poorer outcomes, while higher socioeconomic status (β = 0.17, *p* = 0.014) and employment (β = 0.15, *p* = 0.019) contributed positively. Increasing age, higher BMI, and medication use were also negatively linked to QoL, though to a lesser extent.

Table 2 reports all variables included in the regression model, including beta coefficients, standard errors, confidence intervals, variance inflation factors, and p-values. Variables such as smoking, employment, BMI, and medication use— although listed descriptively in Table 1—were incorporated into this model only for QoL and psychological outcomes, not functional recovery, due to their theoretical and empirical relevance to mental and general health rather than physical rehabilitation alone.

Physical activity demonstrated significant positive correlations with multiple domains of QoL, including physical functioning (r = 0.46, *p* = 0.009), mental health (r = 0.35, *p* = 0.027), vitality (r = 0.31, *p* = 0.038), social functioning (r = 0.29, *p* = 0.041), and general health (r = 0.33, *p* = 0.034), indicating moderate associations with both physical and psychosocial well-being (Table 3). In contrast, negative correlations were observed for anxiety (r = −0.41, *p* = 0.013) and depression (r = −0.38, *p* = 0.022), suggesting that higher levels of physical activity are linked to reduced psychological distress.

The moderation analysis of demographic and lifestyle factors revealed a significant association between physical activity levels and functional recovery outcomes. However, age (β = −0.12, p = 0.037) and the presence of comorbidities (β = −0.21, p = 0.016) negatively moderated this relationship (Fig 4). Gender (β = 0.14, p = 0.027) and socioeconomic status (β = 0.09, p = 0.044) also influenced the moderation effect, with significant interaction terms observed for physical activity with age (β = 0.18, p = 0.011), gender (β = −0.06, p = 0.049), and socioeconomic status (β = 0.11, p = 0.038). Incorporating these interaction terms in the hierarchical regression model significantly enhanced its explanatory power (ΔR² = 0.16, p = 0.023), raising the total R² from 0.52 to 0.68. Interaction terms included physical activity × age (β = 0.18, SE = 0.06, p = 0.011), physical activity × gender (β = −0.06, SE = 0.03, p = 0.049), and physical activity × socioeconomic status (β = 0.11,

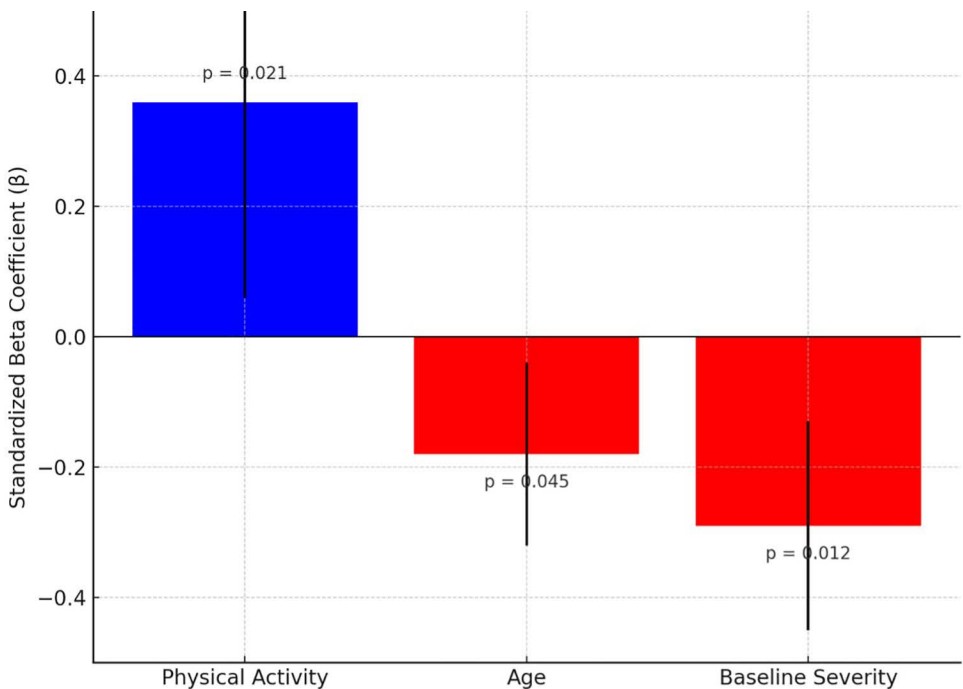

**Fig 2. Standardized beta coefficients with 95% confidence intervals for predictors of functional recovery (n = 286).**

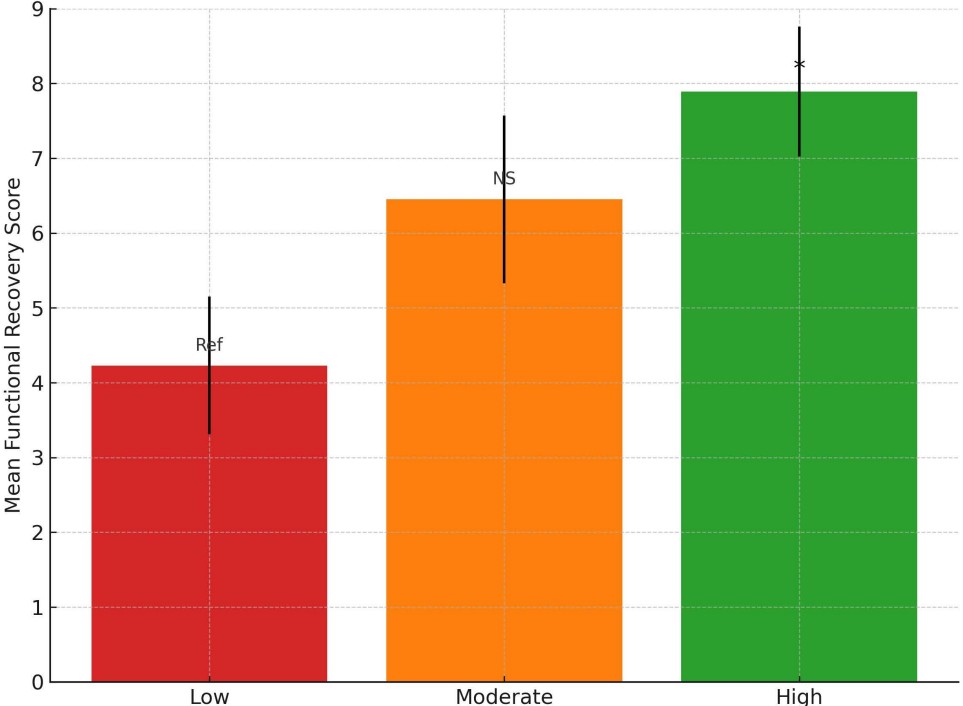

**Fig 3. Mean functional recovery scores by physical activity group (low, moderate, high), with 95% confidence intervals.**

**Table 3. Pearson correlation coefficients between physical activity and quality of life and psychological outcomes.**

| Subscale | Pearson Correlation Coefficient (r) | 95% Confidence Interval | p-value |
|---|---|---|---|
| Physical Functioning (SF-36) | 0.46 | 0.22 to 0.65 | .009 |
| Mental Health (SF-36) | 0.35 | 0.11 to 0.53 | .027 |
| Anxiety (HADS) | −0.41 | −0.61 to −0.18 | .013 |
| Depression (HADS) | −0.38 | −0.56 to −0.14 | .022 |
| Vitality (SF-36) | 0.31 | 0.07 to 0.50 | .038 |
| Social Functioning (SF-36) | 0.29 | 0.06 to 0.48 | .041 |
| General Health (SF-36) | 0.33 | 0.10 to 0.52 | .034 |

r; Pearson Correlation Coefficient; SD; Standard Deviation; SF-36; Short Form Health Survey; HADS; Hospital Anxiety and Depression Scale.

SE $= 0.05$, $p = 0.038$). The overall model after including interactions yielded an $F_{(10, 275)} = 9.24$, $p < 0.001$, with an adjusted $R^2 = 0.68$.

## Discussion

This study explored how physical activity levels relate to functional recovery, QoL, and psychological well-being, while also assessing the moderating roles of demographic and lifestyle factors. In this sample, higher physical activity levels were associated with more favorable recovery outcomes, though these associations may not extend universally to all clinical populations. Linear regression analysis confirmed that physical activity contributed positively to recovery, even after adjusting for factors like age and baseline severity. Furthermore, higher activity levels correlated with enhanced QoL and

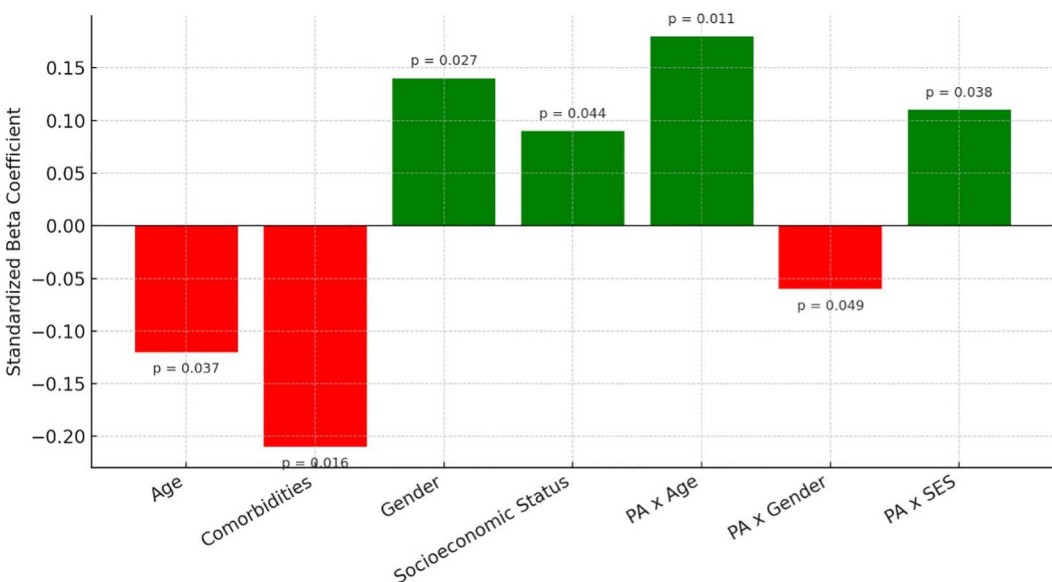

**Fig 4. Moderation plots illustrating interaction effects of age, gender, socioeconomic status, and comorbidities on the relationship between physical activity and functional recovery.**

lower anxiety and depression scores. Moderation analysis indicated that age, gender, socioeconomic status, and comorbidities influenced these relationships, with interaction terms strengthening the model's explanatory power.

Our findings reinforce previously reported associations between physical activity and better functional recovery outcomes in musculoskeletal rehabilitation populations [10]. The significant differences observed in functional recovery between physical activity groups, as demonstrated by the ANOVA and post-hoc tests, highlight a pattern consistent with a dose-related association; however, this should not be interpreted as evidence of a causal dose–response relationship due to the cross-sectional nature of the study. These findings are consistent with previous research in the field [11,12]. Trulsson et al. [13] demonstrated that structured physical activity interventions led to significant improvements in functional recovery among adults with chronic musculoskeletal conditions undergoing physiotherapy. Similarly, studies by Billot et al. [12] and Ginis et al. [14] have shown that regular physical activity is associated with better outcomes in terms of mobility and recovery, particularly in populations with chronic conditions [14]. The negative impact of age and baseline severity on recovery is also well-documented, with Stessel et al. [15] noting that older adults and those with more severe baseline conditions tend to exhibit slower recovery trajectories [15]. The current results align with these studies, reinforcing the critical role of physical activity in enhancing physiotherapy outcomes.

The findings from this study highlight that higher physical activity levels are significantly associated with better QoL and psychological well-being [16]. This could be attributed to several factors [16]. Physically active individuals tend to experience improved cardiovascular and musculoskeletal health, which in turn enhances physical functioning and overall vitality [17]. Additionally, the mental health benefits of physical activity are well-documented, with physical activity promoting the release of endorphins and other neurotransmitters that positively affect mood and reduce anxiety and depression [4]. The negative associations observed with baseline health status and medication use are consistent with the fact that individuals with poorer health conditions or those on medications may experience more limitations, impacting both physical and psychological outcomes [18]. The ANCOVA analysis further supports the importance of physical activity, as higher activity levels were associated with better outcomes across all QoL measures [19]. These results are in line with existing literature. Lucini et al. [20] emphasized that physical activity improves both physical and mental well-being, particularly in

chronic disease populations [20]. Similarly, Wei et al. [21] found that regular physical activity significantly enhances QoL and reduces psychological distress in adults [21]. The negative impact of baseline health status is also supported by the work of Knowles et al. [22] and Cheng et al. [23], who noted that individuals with chronic health issues tend to report lower QoL and greater psychological distress [23]. Moreover, the significant differences across physical activity groups align with the findings of Violant-Holz [24] and Carriedo [25], who demonstrated that higher physical activity levels result in better psychological outcomes.

The moderation analysis revealed that demographic and lifestyle factors, such as age, gender, socioeconomic status, and comorbidities, significantly influenced the relationship between physical activity levels and functional recovery outcomes [26]. Specifically, older age and the presence of comorbidities were associated with reduced functional recovery, likely due to the decreased physical resilience and greater health challenges faced by these individuals [27]. Gender and socioeconomic status also played moderating roles, with higher recovery outcomes observed in males and those with higher socioeconomic status [28]. This could be explained by differences in access to healthcare resources, support systems, and physical therapy adherence linked to socioeconomic gradients, as highlighted in recent musculoskeletal rehabilitation studies [29]. The interaction effects between physical activity and these demographic factors underscore the complexity of recovery, where individual characteristics significantly shape the outcomes of rehabilitation efforts [30]. The hierarchical regression model, which demonstrated improved explanatory power when these interactions were included, further emphasizes the importance of considering these moderating factors in understanding recovery trajectories [31]. These findings are consistent with previous research on the role of demographic and lifestyle factors in rehabilitation outcomes [31]. The study by Izquierdo et al. [32] found that older adults and those with comorbid conditions often experience slower recovery and greater physical limitations, which aligns with the current study's findings of negative associations between age, comorbidities, and recovery [32]. Similarly, Nielsen et al. [33] noted that gender and socioeconomic status play significant roles in health outcomes, with males and individuals from higher socioeconomic groups typically showing better recovery due to increased physical activity participation and access to resources [33]. The interaction between physical activity and age observed in this study mirrors the findings of Dimitri et al. [34], who demonstrated that physical activity has a more substantial positive effect on recovery in younger populations [34]. The interaction between physical activity and age observed in this study mirrors findings in musculoskeletal cohorts, where younger adults benefit more substantially from physical rehabilitation interventions than older adults with comorbidities [35]. These studies collectively support the conclusion that demographic and lifestyle factors significantly moderate the effects of physical activity on functional recovery.

Due to the cross-sectional design of this study, all findings should be interpreted as associations rather than causal effects. Although statistical models revealed significant relationships between physical activity levels and outcome measures, these do not establish temporal sequence or directionality. The observed trends may reflect underlying behavioral, demographic, or clinical characteristics not fully captured in the analysis.

## Clinical significance of the study

The clinical significance of this study lies in demonstrating that higher levels of physical activity are strongly associated with better functional recovery, improved QoL, and enhanced psychological well-being in individuals undergoing physiotherapy for musculoskeletal disorders. Notably, the study highlights that demographic and lifestyle factors, such as age, gender, socioeconomic status, and comorbidities, significantly influence these outcomes. These findings suggest that tailoring physiotherapy treatments to account for a person's demographic and health background—especially in older adults and those with multiple comorbidities—can lead to better recovery results. Furthermore, the positive impact of physical activity underscores the importance of integrating structured physical activity programs as a key component of rehabilitation. Clinicians should consider these moderating factors when designing personalized treatment plans, ensuring to address obstacles to physical activity, such as socioeconomic challenges or health issues, in order to optimize patient recovery and overall well-being.

 

## Limitations of the study

Reliance on self-reported physical activity data introduces several limitations. First, recall bias may affect the accuracy of responses, particularly in estimating frequency and duration of physical activity over a typical week. Second, social desirability bias may lead participants to overreport their physical activity levels to align with perceived normative expectations, potentially inflating MET-min/week values recorded through the IPAQ-SF. Third, psychological status itself may influence self-report accuracy; individuals experiencing anxiety or depression may underreport activity due to negative self-perception or motivational deficits, while others may overreport as a form of compensation or perceived compliance. These factors introduce measurement error that may attenuate or distort observed associations between physical activity and health outcomes. Although weekly activity logs and trained data collectors were employed to mitigate these risks, the inherent limitations of self-report data should be considered when interpreting the strength and direction of the study's findings. Although validated questionnaires were used, objective tools like accelerometers or pedometers would yield more precise data and lessen dependence on memory. Additionally, the study did not incorporate objective severity scores such as the numeric rating scale for pain, which may have provided further granularity in assessing baseline condition severity. Future studies should include such metrics to enhance clinical characterization. Future studies could incorporate these devices or weekly activity logs to improve data accuracy. Additionally, the cross-sectional design restricts causal interpretations of how physical activity affects recovery, indicating that longitudinal research is necessary to track changes over time and determine causation.

## Conclusion

This study concludes that higher levels of physical activity are *associated* with greater functional recovery scores, higher quality of life ratings, and reduced psychological distress among patients undergoing physiotherapy for musculoskeletal disorders. Additionally, demographic and lifestyle factors—such as age, gender, socioeconomic status, and comorbidities—moderate these relationships and influence recovery outcomes. These findings emphasize the relevance of considering physical activity levels as a potential correlate of improved rehabilitation-related outcomes; however, causality cannot be inferred, and future longitudinal studies are needed to determine the direction and mechanisms of these associations.

## Author contributions

**Conceptualization:** Batool Abdulelah Alkhamis, Ravi Shankar Reddy, Mastour Saeed Alshahrani, Zuhair Al Salim, Faisal M. Alyazedi, Ahmad Mohamed Elshiwi, Ghada Mohamed Koura, Devika Rani Sangadala, Debjani Mukherjee, Saeed Y. Al Adal, Hussain Saleh H Ghulam.

**Data curation:** Batool Abdulelah Alkhamis, Ravi Shankar Reddy, Mastour Saeed Alshahrani, Zuhair Al Salim, Faisal M. Alyazedi, Ahmad Mohamed Elshiwi, Ghada Mohamed Koura, Devika Rani Sangadala, Debjani Mukherjee, Saeed Y. Al Adal, Hussain Saleh H Ghulam.

**Formal analysis:** Batool Abdulelah Alkhamis, Ravi Shankar Reddy, Mastour Saeed Alshahrani, Zuhair Al Salim, Faisal M. Alyazedi, Ahmad Mohamed Elshiwi, Ghada Mohamed Koura, Devika Rani Sangadala, Debjani Mukherjee, Hussain Saleh H Ghulam.

**Funding acquisition:** Ravi Shankar Reddy.

**Investigation:** Batool Abdulelah Alkhamis.

**Methodology:** Ravi Shankar Reddy, Mastour Saeed Alshahrani, Zuhair Al Salim, Faisal M. Alyazedi, Ahmad Mohamed Elshiwi, Ghada Mohamed Koura, Devika Rani Sangadala, Debjani Mukherjee, Saeed Y. Al Adal.

**Project administration:** Batool Abdulelah Alkhamis, Ravi Shankar Reddy.

**Resources:** Batool Abdulelah Alkhamis.

**Software:** Batool Abdulelah Alkhamis, Ravi Shankar Reddy, Hussain Saleh H Ghulam.

**Supervision:** Batool Abdulelah Alkhamis, Faisal M. Alyazedi, Ahmad Mohamed Elshiwi, Ghada Mohamed Koura, Saeed Y. Al Adal.

**Validation:** Devika Rani Sangadala.

**Visualization:** Mastour Saeed Alshahrani, Hussain Saleh H Ghulam.

**Writing – original draft:** Batool Abdulelah Alkhamis, Ravi Shankar Reddy, Mastour Saeed Alshahrani, Zuhair Al Salim, Faisal M. Alyazedi, Ghada Mohamed Koura, Saeed Y. Al Adal.

**Writing – review & editing:** Batool Abdulelah Alkhamis, Ravi Shankar Reddy, Mastour Saeed Alshahrani, Faisal M. Alyazedi, Ahmad Mohamed Elshiwi, Ghada Mohamed Koura, Devika Rani Sangadala, Debjani Mukherjee, Saeed Y. Al Adal, Hussain Saleh H Ghulam.

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
