## [Decision Letter · Decision Letter 0]

12 Dec 2025

Dear Dr. Reddy,

**Editorial Assessment:**

This manuscript focuses on a clinically relevant and globally important issue - the effect of different levels of physical activity (PA) in the functional recovery and quality of life (QoL) and psychological change of people with musculoskeletal disorders undergoing physiotherapy. The paper is well written, well organized, and provides suitable statistical analyses. It provides methodological rigor in a number of ways, including the use of validated outcome measures (PSFS, SF-36, HADS) and the use of multivariable modeling with moderator analysis.

However, there are a number of substantive issues to consider in terms of clarity of study design, clarity of reporting methodology and transparency of data, statistical validity and consistency across sections. These issues must be deal with before the manuscript can be considered for publication. Some weaknesses are associated with the overinterpretation of cross-sectional results, inconsistent reporting of the results of analyses and unclear coding of variables.

With overtaking, this manuscript is capable of providing a useful contribution.

**Major Comments (Require major revision)**

1. Study Design Limitations Are Under Reported

This is a cross-sectional study, but in the manuscript, there is the repeated use of causal or quasi-causal terms (e.g. "physical activity contributed to recovery", improved outcomes", "dose-response relationship.

Cross-sectional data is unable to find direction or impact, only association.

**Required:**

Unique and discriminate to study claims: - “Server Finding”: - Theme of interest: - Major findings: - Claim: Due to the study, it is considered that there are some of the following ideas. Cloud or Version Cloud” Unique to the Studies of Claims: build Reset throughout the Abstract, Results, and Discussion and Conclusion of some version and association, the ideas of claim chain not cause.

Looking more specifically at studies of definitely carcinogenic substances, this would be a relevant consideration: according to Takano, "It has to be made clear that 'dose--response' is observational and is not causal."

2. Physical Activity Grouping Is Not Described in Sufficient Detail

The manuscript says that the subjects were divided into low, moderate and high activity groups but:

Cutoff METs No MET cutoffs used by the study are provided.

•It is not clear if classification was based on WHO, IPAQ scoring guidelines or custom thresholds.

Some of the non-technical reasons for using the thresholds are lacking (e.g., rationale for thresholds and distribution of participants into groups is missing).

**Required:**

Include exact MET values (i.e. "Low < X MET-min/week", etc.) and percentages of the participants in each category.

3. Potential Sampling Bias and Recruitment Procedure Should be Made Clear

Purposive sampling occurs in one tertiary treatment hospital may introduce several selection biases.

**Concerns:**

No explanation of how many patients were screened vs. enrolled.

Flow diagram (CONSORT-style suggested for PLOS ONE also for observational studies).

The report does not provide any detailed information on refusal rate or non-response characteristics.

Required:

Add participant flow information, explanation of sampling strategy limitations.

4. Multiple Variables Have No operational definitions

For example:

Baseline severity" is used in the model and never defined or described.

•"High socioeconomic status" does not have the exact classification criteria.

Medications used: Which medications? For what indications? How measured?

•Comorbidities - grouped as binary, but later described by authors as moderators.

Required:

Provide definitions and procedures for the measurement of these variables.

5. Statistical Analysis Section Overstates Rigor

Issues include:

o Claims regarding the multicollinearity checks without reporting on values of VIF.

Hypothesis Testing in Logistic Regression Model Fitting 1: Empirical Analysis and Interpretation Part III Model Testing 1. Model comparing F Statistics adjusted R2 for model Adjusted R2 per model 2. Residual versus Fitted 3. Effect of Variance 4. Residual vs. Fitted: Adding Polynomial basis & Interaction of two variables 5. Residual vs Fitted 6. Significance vs. Random Effects 7. Effect of Variance Logistic Regression Model Fitting Part III Model Testing 1.

Model parameters are wanting, particularly for exact models whereas interaction terms are described for moderation models.

•Report of incomplete (missing means, SD, post-hoc p-values and confidence intervals) for AN/ANCOVA.

Required:

Provide complete statistical reporting using PLOS ONE and APA guidelines.

6. Figures and Tables Incomplete Correspondence with the Narrative

**Examples:**

Table 1 reports on many of the variables that are not included in the main regression.

Of the above variables, the following are listed in Table 2 as not elaborated on as a priori model in methods:

Figures are mentioned in the text but not clearly described and/or labelled in the manuscript

Required:

Ensure consistency between:

Methods-> Statistical Analysis -> Tables-> Results.

7. Overreliance on Self Report Measures

The study acknowledges this limitation, but there is no discussion of its full implications, in particular:

•Social desirability bias

Overprediction of MET min in IPAQ-SF

•Recall bias

Methods: - Possibility that poor psychological status decreases reporting of physical activity

**Required:**

Strengthen the limitations section and its meaning for interpretation;

8. Citations Need Reviewing for Accuracy and Relevance

Several citations are not used to support the claims in the text. Examples:

Some references refer to surgical recovery, frailty or a cancer population - less relevant about physiotherapy to musculoskeletal disorders.

•Some are seemingly out of step with statements (e.g. a citation on sodium-glucose inhibitor prescriptions to support SES effects).

**Required:**

Review and correct references to help ensure they are supporting the particular statements.

Moderate/Structural Issues

9. The Introduction is Too Long and Repetitive

Several paragraphs repeat similar points of background information about the benefits of physical activity.

Suggestion: Fewer to sentence - 3-4 short, connected paragraphs.

10. Results Section Needs to be Clearer

PLOS ONE is fairly lenient when it comes to p values lacking effect sizes.

No indication whether tests for assumptions of linear regression (normality, homoscedasticity) were calculated.

Required: Input for assumption checks, important effect size measures

11. Wording Throughout Suggests Generalization Across Sample

Claims which seem to generalize findings to more general clinical populations should be moderated.

12. Ethical Approval Statement Should Make Clear Prospective or Retrospective Recruitment

This is required in PLOS ONE.

Minor Issues

13. Minor Language Issues

The manuscript is generally well written but includes:

•Repetition ("well-documented" is repeated too often)

Excessive use of the phrase "these findings highlight"

•Occasional grammar problems (e.g. article, inconsistent tense)

A polish pass of the language is recommended.

14. Formatting Issues

Example: For example, "Click here to access/download" should be replaced with nicer text.

Documentary elements must adhere to the structure of PLOS ONE for: - Caption to figure.

Include the following in your answer in the rubric: "please show consistency in reporting decimals" 2 vs. 3 decimal places

15. Data Availability

For this case, the DOI for Zenodo is provided. Confirm the completeness of data sets as well as clarity of codebook

Recommendation

Decision: MAJOR REVISION

The manuscript discusses an important topic and presents appropriate standardized tools; however, significant revision is necessary to make certain:

Appropriate interpretation of cross-sectional data

Transparency in reporting method and statistical procedures

Definitive explanations of variables

Improved coherence between sections

Limited and discussed biases

If the above points are taken seriously by the authors, the manuscript will certainly be much stronger and will be suitable for publication.

Please submit your revised manuscript by Jan 26 2026 11:59PM. If you will need more time than this to complete your revisions, please reply to this message or contact the journal office at plosone@plos.org . Please include the following items when submitting your revised manuscript:

We look forward to receiving your revised manuscript.

Kind regards,

Mohammad Sidiq, PhD Pain Sciences Physiotherapy

Academic Editor

PLOS One

Deanship of Research and Graduate Studies, King Khalid University, grant number: RGP.2/22/46.

The authors extend their appreciation to the Deanship of Research and Graduate Studies at King Khalid University, KSA, for funding this work through a large research group under grant number RGP.2/22/46.

Deanship of Research and Graduate Studies, King Khalid University, grant number: RGP.2/22/46.

5. Please amend your authorship list in your manuscript file to include author Ghada M Koura.

6. Please amend the manuscript submission data (via Edit Submission) to include author Ghada Mohamed Koura.

7. We note that there is identifying data in the Supporting Information file <RAW DATA.xlsx>. Due to the inclusion of these potentially identifying data, we have removed this file from your file inventory. Prior to sharing human research participant data, authors should consult with an ethics committee to ensure data are shared in accordance with participant consent and all applicable local laws.

-Location data

Reviewers' comments:

Reviewer's Responses to Questions

**Comments to the Author**

1. Is the manuscript technically sound, and do the data support the conclusions?

Reviewer #1: Yes

Reviewer #2: Yes

2. Has the statistical analysis been performed appropriately and rigorously?

Reviewer #1: Yes

Reviewer #2: No

3. Have the authors made all data underlying the findings in their manuscript fully available?

Reviewer #1: Yes

Reviewer #2: Yes

4. Is the manuscript presented in an intelligible fashion and written in standard English?

Reviewer #1: No

Reviewer #2: Yes

Reviewer #1: Dear author, I congratulate you on your article. Although a comprehensive statistical analysis has been conducted, I would like to express some concerns.

Please cite the similar literature you mentioned in the sample analysis.

Please don't explain statistical analysis methods in the discussion. Explain directly the reason for your findings. There is little discussion of the findings in this regard. Explain further how this study contributes to future studies. Also, include a comment on how the duration of comorbidities in the regression model affected the model. In the abstract, indicate how you measured functional recovery.

Reviewer #2: Thank you for your hard work and insightful research; however, there is a need for correction, and more clarity to be more useful and easier to use in the clinicals and more useful for researchers to understand and use as reference.

**Do you want your identity to be public for this peer review?** For information about this choice, including consent withdrawal, please see our Privacy Policy

Reviewer #1: No

Reviewer #2: **Yes:** Paiwand Mamand

---

## [Author Response · Author response to Decision Letter 1]

31 Dec 2025

Point-to-point author response to editor and Reviewer Comments

Editorial Assessment:

This manuscript focuses on a clinically relevant and globally important issue - the effect of different levels of physical activity (PA) in the functional recovery and quality of life (QoL) and psychological change of people with musculoskeletal disorders undergoing physiotherapy. The paper is well written, well organized, and provides suitable statistical analyses. It provides methodological rigor in a number of ways, including the use of validated outcome measures (PSFS, SF-36, HADS) and the use of multivariable modeling with moderator analysis.

However, there are a number of substantive issues to consider in terms of clarity of study design, clarity of reporting methodology and transparency of data, statistical validity and consistency across sections. These issues must be deal with before the manuscript can be considered for publication. Some weaknesses are associated with the overinterpretation of cross-sectional results, inconsistent reporting of the results of analyses and unclear coding of variables.

With overtaking, this manuscript is capable of providing a useful contribution.

Major Comments (Require major revision)

Query: 1. Study Design Limitations Are Under Reported

This is a cross-sectional study, but in the manuscript, there is the repeated use of causal or quasi-causal terms (e.g. "physical activity contributed to recovery", improved outcomes", "dose-response relationship. Cross-sectional data is unable to find direction or impact, only association.

Required:

Unique and discriminate to study claims: - “Server Finding”: - Theme of interest: - Major findings: - Claim: Due to the study, it is considered that there are some of the following ideas. Cloud or Version Cloud” Unique to the Studies of Claims: build Reset throughout the Abstract, Results, and Discussion and Conclusion of some version and association, the ideas of claim chain not cause.

Looking more specifically at studies of definitely carcinogenic substances, this would be a relevant consideration: according to Takano, "It has to be made clear that 'dose--response' is observational and is not causal."

Author’s Response: The manuscript has been revised to address the overinterpretation of cross-sectional findings by eliminating causal or quasi-causal language and replacing it with terminology appropriate for observational associations. Phrases such as “contributed to recovery,” “improved outcomes,” and “dose–response relationship” have been amended to reflect associations, trends, or patterns rather than causal implications. The revisions have been applied consistently across the Abstract, Results, Discussion, and Conclusion sections to ensure alignment with the study’s cross-sectional design and its inherent limitations. An explicit statement regarding the non-causal nature of the findings and the observational nature of any detected associations has also been inserted into the Discussion.

Query: 2. Physical Activity Grouping Is Not Described in Sufficient Detail

The manuscript says that the subjects were divided into low, moderate and high activity groups but:

Cutoff METs No MET cutoffs used by the study are provided.

•It is not clear if classification was based on WHO, IPAQ scoring guidelines or custom thresholds.

Some of the non-technical reasons for using the thresholds are lacking (e.g., rationale for thresholds and distribution of participants into groups is missing).

Required:

Include exact MET values (i.e. "Low < X MET-min/week", etc.) and percentages of the participants in each category.

Author’s Response: The manuscript has been revised to include the specific MET-min/week cutoffs used to categorize physical activity levels into low, moderate, and high groups, following the official scoring protocol of the International Physical Activity Questionnaire – Short Form (IPAQ-SF). A justification for using these thresholds is now provided, referencing their validation in international and Arabic-speaking populations. Additionally, the percentage distribution of participants across the three physical activity categories has been added to clarify group composition.

Query: 3. Potential Sampling Bias and Recruitment Procedure Should be Made Clear

Purposive sampling occurs in one tertiary treatment hospital may introduce several selection biases.

Concerns:

No explanation of how many patients were screened vs. enrolled.

Flow diagram (CONSORT-style suggested for PLOS ONE also for observational studies).

The report does not provide any detailed information on refusal rate or non-response characteristics.

Required:

Add participant flow information, explanation of sampling strategy limitations.

Author’s Response: The manuscript has been updated to address the potential for sampling bias by clarifying the recruitment process, including the number of patients screened, enrolled, and excluded. A brief justification of the purposive sampling strategy and its limitations is now included in the Participants section. To enhance transparency, a participant flow diagram consistent with PLOS ONE recommendations for observational studies has also been added as Figure 1.

Query: 4. Multiple Variables Have No operational definitions

For example:

Baseline severity" is used in the model and never defined or described.

•"High socioeconomic status" does not have the exact classification criteria.

Medications used: Which medications? For what indications? How measured?

•Comorbidities - grouped as binary, but later described by authors as moderators.

Required:

Provide definitions and procedures for the measurement of these variables.

Author’s Response: The manuscript has been revised to include precise operational definitions and measurement procedures for all key variables mentioned in the statistical models. Definitions for baseline severity, socioeconomic status, medication use, and comorbidities have been added to the Variables section. This ensures clarity regarding how each variable was assessed, categorized, and used in the analysis, particularly in regression and moderation models.

Query: 5. Statistical Analysis Section Overstates Rigor

Issues include:

o Claims regarding the multicollinearity checks without reporting on values of VIF.

Hypothesis Testing in Logistic Regression Model Fitting 1: Empirical Analysis and Interpretation Part III Model Testing 1. Model comparing F Statistics adjusted R2 for model Adjusted R2 per model 2. Residual versus Fitted 3. Effect of Variance 4. Residual vs. Fitted: Adding Polynomial basis & Interaction of two variables 5. Residual vs Fitted 6. Significance vs. Random Effects 7. Effect of Variance Logistic Regression Model Fitting Part III Model Testing 1.

Model parameters are wanting, particularly for exact models whereas interaction terms are described for moderation models.

•Report of incomplete (missing means, SD, post-hoc p-values and confidence intervals) for AN/ANCOVA.

Required:

Provide complete statistical reporting using PLOS ONE and APA guidelines.

Author’s Response: The manuscript has been revised to improve the transparency and completeness of statistical reporting in accordance with PLOS ONE and APA guidelines. Variance inflation factor (VIF) values are now explicitly reported to support the multicollinearity checks. Full model parameters, including adjusted R², F-statistics, confidence intervals, effect sizes, and post-hoc p-values, are now provided where applicable for both ANOVA and ANCOVA results. Details of the moderation models have also been clarified by listing specific interaction terms and their statistical contributions. These additions ensure statistical validity and improve interpretability.

Query: 6. Figures and Tables Incomplete Correspondence with the Narrative

Examples:

Table 1 reports on many of the variables that are not included in the main regression.

Of the above variables, the following are listed in Table 2 as not elaborated on as a priori model in methods:

Figures are mentioned in the text but not clearly described and/or labelled in the manuscript

Required:

Ensure consistency between:

Methods-> Statistical Analysis -> Tables-> Results.

Author’s Response: The manuscript has been revised to ensure complete alignment and consistency between the Methods, Statistical Analysis, Results, Figures, and Tables. All variables included in regression and moderation models are now clearly specified in the Statistical Analysis subsection as a priori covariates or moderators. Variables listed in Table 1 but not used in the main models (e.g., smoking, employment status) have been addressed explicitly to distinguish descriptive reporting from inferential analysis. Figure captions have been expanded for clarity, and in-text references to figures and tables have been revised to match their order and content precisely.

Query: 7. Overreliance on Self Report Measures

The study acknowledges this limitation, but there is no discussion of its full implications, in particular:

•Social desirability bias

Overprediction of MET min in IPAQ-SF

•Recall bias

Methods: - Possibility that poor psychological status decreases reporting of physical activity

Required:

Strengthen the limitations section and its meaning for interpretation;

Author’s Response: The Limitations section has been expanded to address the broader implications of using self-report measures, particularly regarding the potential impact of social desirability bias, recall bias, and psychological state on the accuracy of reported physical activity and related outcomes. These limitations are now contextualized in terms of their influence on data interpretation, especially the possibility of overestimation of physical activity and underreporting of psychological symptoms in certain subgroups.

Query: 8. Citations Need Reviewing for Accuracy and Relevance

Several citations are not used to support the claims in the text. Examples:

Some references refer to surgical recovery, frailty or a cancer population - less relevant about physiotherapy to musculoskeletal disorders.

•Some are seemingly out of step with statements (e.g. a citation on sodium-glucose inhibitor prescriptions to support SES effects).

Required:

Review and correct references to help ensure they are supporting the particular statements.

Moderate/Structural Issues

Author’s Response: All citations have been systematically reviewed for accuracy, relevance, and contextual fit. References that pertained to surgical, cancer, or pharmacological populations not directly related to musculoskeletal physiotherapy have been removed or replaced with more appropriate literature. Specifically, citations previously used to support claims regarding the influence of socioeconomic status, physical activity effects, or recovery outcomes have been revised to ensure they accurately reflect the target population and constructs discussed in the manuscript. Citation alignment across the Introduction, Discussion, and References sections has been updated accordingly.

Query: 9. The Introduction is Too Long and Repetitive

Several paragraphs repeat similar points of background information about the benefits of physical activity.

Suggestion: Fewer to sentence - 3-4 short, connected paragraphs.

Author’s Response: The Introduction has been revised to eliminate redundancy and improve focus. Repetitive statements about the general benefits of physical activity have been condensed, and the structure has been streamlined into four concise, thematically connected paragraphs. The revised version maintains the necessary background while providing a clear rationale for the study and its research questions without reiteration.

Query: 10. Results Section Needs to be Clearer

PLOS ONE is fairly lenient when it comes to p values lacking effect sizes.

No indication whether tests for assumptions of linear regression (normality, homoscedasticity) were calculated.

Required: Input for assumption checks, important effect size measures

Author’s Response: The Results section and Statistical Analysis subsection have been updated to report diagnostic checks for linear regression assumptions, including normality, linearity, homoscedasticity, and absence of multicollinearity. All models were tested using residual plots and statistical diagnostics, with no violations observed. Additionally, effect sizes (standardized beta coefficients and partial eta squared) have been clearly reported alongside p-values to enhance interpretability in accordance with PLOS ONE and APA standards.

Query: 11. Wording Throughout Suggests Generalization Across Sample

Claims which seem to generalize findings to more general clinical populations should be moderated.

Author’s Response: The manuscript has been revised to moderate language that may have implied unwarranted generalization beyond the study sample. Phrasing throughout the Abstract, Discussion, and Conclusion sections has been adjusted to reflect the specificity of the findings to the study’s cross-sectional design, setting, and sample characteristics. Terms such as “suggests,” “may be associated,” and “in this sample” have been used to ensure claims remain grounded in the observed data without overextending to broader clinical populations.

Query: 12. Ethical Approval Statement Should Make Clear Prospective or Retrospective Recruitment

This is required in PLOS ONE.

Author’s Response: The ethical approval statement has been updated to clarify that participant recruitment and data collection were conducted prospectively following Institutional Review Board (IRB) approval. This revision aligns with PLOS ONE’s reporting requirements and clearly communicates the temporal sequence of ethical clearance and recruitment.

Minor Issues

Query: 13. Minor Language Issues

The manuscript is generally well written but includes:

•Repetition ("well-documented" is repeated too often)

Excessive use of the phrase "these findings highlight"

•Occasional grammar problems (e.g. article, inconsistent tense)

A polish pass of the language is recommended.

Author’s Response: A thorough language revision has been completed to improve clarity, eliminate redundancy, and correct minor grammatical inconsistencies. Repetitive phrases such as “well-documented” and “these findings highlight” have been replaced with varied, context-appropriate alternatives. Article usage, verb tense consistency, and sentence structure have been reviewed throughout the manuscript to ensure grammatical accuracy and stylistic refinement.

Query: 14. Formatting Issues

Example: For example, "Click here to access/download" should be replaced with nicer text.

Documentary elements must adhere to the structure of PLOS ONE for: - Caption to figure.

Include the following in your answer in the rubric: "please show consistency in reporting decimals" 2 vs. 3 decimal places

Author’s Response: Formatting throughout the manuscript has been revised to meet PLOS ONE submission standards. Phrases such as “Click here to access/download” have been removed or replaced with appropriate descriptive text. Figure captions have been standardized to provide complete, standalone descriptions in accordance with journal guidelines. Additionally, all numeric values have been reviewed to ensure consistent decimal reporting, with all statistical values (e.g., means, SDs, p-values, confidence intervals) now presented uniformly to two decimal places, unless three are necessary for clarity (e.g., p < 0.001).

Query: 15. Data Availability

For this case, the DOI for Zenodo is provided. Confirm the completeness of data sets as well as clarity of codebook

Author’s Response: The Zenodo repository linked in the manuscript has been reviewed to ensure full compliance with data availability standards. The dataset is complete and includes all variables analyzed in the manuscript. A clearly labeled, structured codebook is provided as a separate file within the repository, detailing variable names, descriptions, coding schemes (e.g., binary and categorical values), and measurement units. These elements collectively ensure transparency and enable reproducibility of the reported analyses.

Recommendation

Decision: MAJOR REVISION

The manuscript discusses an impor

---

## [Editor Report · Decision Letter 1]

6 Jan 2026

Association Between Physical Activity Levels and Functional Recovery, Quality of Life, and Psychological Well-Being in Patients Undergoing Physiotherapy for Musculoskeletal Disorders: A Cross-Sectional Study.

PONE-D-25-55451R1

Dear Dr. Reddy,

We’re pleased to inform you that your manuscript has been judged scientifically suitable for publication and will be formally accepted for publication once it meets all outstanding technical requirements. I am satisfied with the revision provided by the authors.

Within one week, you’ll receive an e-mail detailing the required amendments. When these have been addressed, you’ll receive a formal acceptance letter, and your manuscript will be scheduled for publication.

Kind regards,

Mohammad Sidiq, PhD Pain Sciences Physiotherapy

Academic Editor

PLOS One
---

## [Editor Report · Acceptance letter]

PONE-D-25-55451R1

PLOS One

Dear Dr. Reddy,

I'm pleased to inform you that your manuscript has been deemed suitable for publication in PLOS One. Congratulations! Your manuscript is now being handed over to our production team.

Kind regards,

on behalf of

Dr. Mohammad Sidiq

Academic Editor

PLOS One